# A multi-stage process to develop quality indicators for community-based palliative care using interRAI data

Dawn M. Guthrie[1,2☯]*, Nicole Williams[1☯], Cheryl Beach[3‡], Emma Buzath[4‡], Joachim Cohen[5‡], Anja Declercq[6,7‡], Kathryn Fisher[8‡], Brant E. Fries[9‡], Donna Goodridge[10‡], Kirsten Hermans[6‡], John P. Hirdes[11‡], Hsien Seow[12‡], Maria Silveira[13‡], Aynharan Sinnarajah[14‡], Susan Stevens[15‡], Peter Tanuseputro[16‡], Deanne Taylor[17,18‡], Christina Vadeboncoeur[19,20,21‡], Tracy Lyn Wityk Martin[4‡]

1 Department of Kinesiology and Physical Education, Wilfrid Laurier University, Waterloo, Ontario, Canada, 2 Department of Health Sciences, Wilfrid Laurier University, Waterloo, Ontario, Canada, 3 Integrated Community Services, Fraser Health, Surrey, British Columbia, Canada, 4 Provincial Palliative and-End-of-Life Care, Seniors Health and Continuing Care, Alberta Health Services, Calgary, Alberta, Canada, 5 End-of-Life Care Research Group, Vrije Universiteit Brussel, Brussels, Belgium, 6 LUCAS – Center for Care Research and Consultancy, KU Leuven, Leuven, Belgium, 7 CESO – Center for Sociological Research, KU Leuven, Leuven, Belgium, 8 School of Nursing, McMaster University, Hamilton, Ontario, Canada, 9 Department of Health Management and Policy and Department of Geriatric and Palliative Medicine, University of Michigan, Ann Arbor, Michigan, United States of America, 10 College of Medicine, University of Saskatchewan, Saskatoon, Saskatchewan, Canada, 11 School of Public Health Sciences, University of Waterloo, Waterloo, Ontario, Canada, 12 Department of Oncology, McMaster University, Hamilton, Ontario, Canada, 13 Division of Geriatric and Palliative Medicine, Department of Internal Medicine, University of Michigan, Ann Arbor, Michigan, United States of America, 14 Department of Medicine, Queen's University, Kingston, Ontario, Canada, 15 Nova Scotia Health, Halifax, Halifax, Nova Scotia, Canada, 16 Department of Medicine, University of Ottawa, Ottawa, Ontario, Canada, 17 Research Department, Interior Health Authority, Kelowna, British Columbia, Canada, 18 Rural Coordination Centre of BC, Penticton, British Columbia, Canada, 19 Department of Pediatrics, University of Ottawa, Ottawa, Ontario, Canada, 20 Children's Hospital of Eastern Ontario Ottawa, Ontario, Canada, 21 Roger Neilson House, Ottawa, Ontario, Canada

☯ These authors contributed equally to this work.
‡ These authors also contributed equally to this work.
* dguthrie@wlu.ca

**Data Availability Statement:** The data can be found at the DOI: https://doi.org/10.5683/SP3/8U8B83.

## Abstract

### Background

Individuals receiving palliative care (PC) are generally thought to prefer to receive care and die in their homes, yet little research has assessed the quality of home- and community-based PC. This project developed a set of valid and reliable quality indicators (QIs) that can be generated using data that are already gathered with interRAI assessments—an internationally validated set of tools commonly used in North America for home care clients. The QIs can serve as decision-support measures to assist providers and decision makers in delivering optimal care to individuals and their families.

### Methods

The development efforts took part in multiple stages, between 2017–2021, including a workshop with clinicians and decision-makers working in PC, qualitative interviews with

**Funding:** DG received an award from the Canadian Institutes of Health Research (CIHR; grant #PJT-156359). CIHR web site:https://cihr-irsc.gc.ca/e/193.html. The funders had no role in study design, data collection and analysis, decision to publish, or preparation of the manuscript.

**Competing interests:** I have read the journal's policy and the authors of this manuscript have the following competing interests: Dr. A. Sinnarajah has received research grants (last 5 years) on palliative care (as principal investigator or co-investigator), from the Canadian Institutes of Health Research, MSI Foundation, Canadian Cancer Society, Applied Research in Cancer Control, College of Family Physicians of Canada, Choosing Wisely Alberta, Alberta Innovates Health Research, Alberta Cancer Foundation, Alberta Health Services, University of Calgary, Canadian Frailty Network, Alberta Health and Campus Alberta Health Outcomes and Public Health. He has an academic appointment for palliative care research with Queen's University and Lakeridge Health (currently), and University of Calgary (last 5 years). Lastly, he is/has paid medical administrative positions with Alberta Health Services and Lakeridge Health. The remaining authors declare that no competing interests exist. We confirm that Dr. Sinnarajah has some competing interests to report (previously detailed in the online submission) and we confirm that this does not alter our adherence to PLOS ONE policies on sharing data and materials.

individuals receiving PC, families and decision makers and a modified Delphi panel, based on the RAND/ULCA appropriateness method.

## Results

Based on the workshop results, and qualitative interviews, a set of 27 candidate QIs were defined. They capture issues such as caregiver burden, pain, breathlessness, falls, constipation, nausea/vomiting and loneliness. These QIs were further evaluated by clinicians/decision makers working in PC, through the modified Delphi panel, and five were removed from further consideration, resulting in 22 QIs.

## Conclusions

Through in-depth and multiple-stakeholder consultations we developed a set of QIs generated with data already collected with interRAI assessments. These indicators provide a feasible basis for quality benchmarking and improvement systems for care providers aiming to optimize PC to individuals and their families.

## Introduction

The goal of palliative care (PC) is to improve quality of life for individuals and their families facing the problems associated with a life-limiting illness, and to provide care that promotes dignity, respect, and comfort [1, 2]. Palliative and end-of-life care spans the disease process from early diagnosis to end-of-life, inclusive of bereavement. A majority of Canadians surveyed in 2016 supported resources being available for PC at home [3]. Recent data from Ontario—one of the sites of this study and the largest province in Canada with more than 14 million residents–found that among the approximately 50% of decedents who received PC, 43% (roughly 54,000 over the course of one year), had palliative home care services [4]. Despite this, little research has assessed the quality of home-based palliative services [5–7], instead focusing on utilization, organization, and cost-effectiveness of these services [8, 9]. Existing QIs in the literature tend to focus on hospital use [5, 10–12]. Currently, Canada does not have a standard set of quality indicators (QIs) for community-based PC. Having valid and reliable QIs, grounded in a community-based perspective, is essential to identify and monitor areas for improvement [13], and ultimately, contribute to the delivery of optimal care to individuals receiving PC. QIs have enormous potential for improving care by providing important information to a variety of users, such as care providers, consumers, accreditation organizations and researchers [14].

QI rates are often used to compare providers, at a systems level, a process known as benchmarking [15]. Benchmarking can also be thought of as a starting point to understand which factors contribute to quality improvement, to promote discussions among health care providers to encourage organizational change within the organizations being compared, and to learn from each other's improvement strategies [16].

Although QIs have been proposed for those receiving PC, or individuals with life-limiting illnesses [17–23], they have several limitations. The majority of existing QIs tend to rely on administrative data to capture hospital admissions/emergency department (ED) visits and focus on the process or structure of care rather than on outcomes that matter most to individuals receiving PC [20, 24–26]. They also tend to be focused predominantly on individuals with

cancer [17, 27, 28], despite the fact that cancer is the cause of death of less a third of decedents, and despite PC being increasingly available to those with other life-limiting conditions such as organ failure and dementia [29]. Therefore, QIs are needed that encompass a wide range of individuals who are receiving PC or who could benefit from such care, including those for whom prognosis may be less predictable [30, 31].

We outline the development of a set of proposed QIs that were explicitly defined based on data elements within the interRAI assessments. interRAI is a not-for-profit network of researchers, clinicians, and policy experts from just over 30 countries who develop and test standardized, clinical assessments for use in a variety of health and social services settings (e.g., home care, nursing homes/long-term care, inpatient mental health) [32–34]. These assessments are extensively used in multiple parts of Canada, and around the world, and the data are available to interRAI researchers and their trainees. For example, the interRAI Home Care assessment (interRAI HC) is currently used in 21 countries [35, 36] and for all long-term home care clients in Ontario. The assessments are performed by trained assessors (typically registered nurses) using all sources of information, including the person receiving care, informal caregivers (natural support persons) and families, clinical care providers and the medical record. The assessment is completed in Ontario for all home care clients who require home care services for a minimum of 60 days.

Utilizing the existing, routinely-collected, and population-based interRAI data to generate QIs is a very efficient use of this information and avoids the tremendous time and energy required using traditional approaches to quality improvement, such as using detailed chart reviews [37]. In fact, the interRAI data represents one of the only databases in Canada with a sufficient number of assessments and unique individuals to allow for the creation and testing of QIs. There is also a long history of QI development and testing by interRAI researchers [14, 38–40], with the earliest QIs proposed in the mid-1990s [41]. Earlier work by our team, using the interRAI assessment for home care, identified a set of preliminary QIs for community-based PC that could be generated with these assessments [42], and also explored the rates of these preliminary QIs by province/territory [43].

The main goal of this paper is to report on the first steps, of a larger study, to develop, test and validate a set of QIs for community-based PC. In this paper, we describe the creation of the QIs that were developed through in-depth consultations with multiple stakeholder groups and evaluation by a panel of experts.

## Materials and methods

QI development took place over approximately four years (2017–2021) and is still ongoing (Fig 1). The research team included 18 researchers, clinicians or decisions makers from Canada, the US and Belgium with expertise in palliative medicine, health services research, epidemiology, and knowledge translation. The team was actively involved in all stages of the development of the QIs, which included three sequential phases. The study protocol was reviewed and approved by the Wilfrid Laurier University Research Ethics Board (REB #5683).

### Phase I: Qualitative input from key stakeholders

The first activity, in this phase, involved input from decision makers from across Ontario, Canada's largest province. A group of 30 PC experts participated in a one-day workshop, the results of which have been published previously [44]. Participants included clinical leaders, researchers, front-line care providers as well as health and information system administrators. At the time of the workshop, the province was divided geographically into 14 Local Health Integration Networks (LHINs) and 12 of the 14 LHINs were represented. Other participants

**2017-2021 Phase I**
- One-day workshop with 30 PC experts from Ontario
- Interviews/focus groups with one patient and several family members and decision makers (n=21) from across Canada

**Phase II**
- Development of QI definitions based on data elements within interRAI instruments
- 27 QIs developed in line with feedback received during Phase I and using the current literature

**Phase III**
- Modified Delphi panel with 21 PC experts from Canada, the US and Belgium
- Panel members evaluated and scored each QI across 4 main criteria
- Scores were then used to decide which QIs were to be retained, reviewed or discarded
- This resulted in 22 QIs which were kept for further consideration

**Fig 1. Summary of the steps in the QI development process.**

represented key organizations such as Hospice Palliative Care Ontario, the Ontario Ministry of Health and Long-Term Care, Ontario Palliative Care Network and the Canadian Institute for Health Information.

In summary, the qualitative analysis of the information provided by the workshop participants identified six key themes important for measuring quality in community-based PC: access to care, patient care, caregiver support, symptom management, spiritual care, and home as the preferred place of death. Where possible, QIs were created that directly related to these six themes (as described below in "Phase II"). For example, since participants discussed symptom management as an important area to assess, we created QIs to measure pain, shortness of breath, fatigue, and other troublesome physical symptoms. The indicator concepts, structure, and definitions were derived from the validated interRAI data elements.

Following the workshop, project Knowledge Users (KUs) helped the research team to recruit individuals receiving PC, their family members and decision makers from across Canada. It was considered inappropriate, and potentially a violation of research ethics, for the research team to directly contact individuals receiving PC and their families. As such, the KUs assisted the team in recruiting potential study participants. KUs represent individuals who are likely to be able to use research results to make informed decisions about health policies, programs and/or practices. The eight KUs on our team included individuals from five different provinces/territories. A series of interviews and focus groups were held with

families (n = 9) and decision makers (n = 11), including one individual who was actively receiving PC. The primary goal of these interviews and focus groups was to understand participants' experiences within the health care system and associate these experiences to measurable QIs that could be developed with existing interRAI data. For example, it was apparent from the interviews that caregivers often felt overwhelmed in their caregiving role and expressed that there was a clear disconnect between what the system could provide and what caregivers expected from the system. Palliative care typically strives to provide access to on-call practitioners during and after office hours. Despite this, some care caregivers felt that during a crisis, they had to rely on ambulances and the use of the emergency department (ED). Caregivers cited emotional and psychological needs as well as loneliness among those receiving PC [45]. This rich qualitative data was used to guide QI development in the next phase of the project.

## Phase II: Defining potential QIs

The research team used the feedback from Phase I, along with current literature [46, 47], to define potential QIs that captured the important domains related to PC quality. For example, based on the workshop results, a QI was drafted to capture the rate of caregiver distress, and other QIs were created to capture issues related to symptom management (e.g., QIs related to pain, breathlessness, falls, constipation, nausea/vomiting). Since the family caregivers and decision makers also discussed issues of accessing the ED, QIs were also developed related to ED visits and hospital admissions. QIs were defined to capture issues such as negative mood, anxiety and loneliness since these were discussed by caregivers and are supported as important for quality assessment in the literature [48, 49].

To generate these QIs, we focused on two interRAI assessment systems, namely the interRAI Home Care instrument (interRAI HC) and the interRAI Palliative Care (interRAI PC) tool, since these are currently used in care planning for home care clients, and palliative clients, in various parts of Canada. The interRAI HC, for example, is completed fully across Ontario, Newfoundland and Labrador and Yukon Territory, with partial coverage in British Columbia and Alberta [50], resulting in roughly 250,000 assessments annually. Research on the interRAI instruments supports the validity and reliability of these data and concludes that the overall quality can be trusted when used to support decision-making [51].

Both of these assessments provide several of the same validated health index scales, which are generated automatically once the assessment is completed. These scales include the Depression Rating Scale (DRS) [52], the Pain Scale [53], the Cognitive Performance Scale [54], the Caregiver Risk Evaluation [55], and the Pressure Ulcer Risk Scale [56]. Since these scales have confirmed validation, the scores on these scales were used in the QI definitions when possible. For example, for the QI on the prevalence of negative mood, the QI definition uses the DRS score of 3+, a cut-point with established predictive validity [52]. Other QIs are based on individual assessment items. Each QI has a distinct numerator and denominator (S1 Table). The QIs fall into two broad categories, namely, "follow-up prevalence" QIs and "failure to improve" QIs (Table 1). The first group captures the prevalence of the issue on re-assessment. For these QIs, the rate was based on re-assessments and admission assessments were excluded from the calculation. This is necessary since admission assessments would not truly reflect quality of PC at the time of the assessment. The "failure to improve" QIs assess the lack of improvement on the issue over two points in time. This is important to capture since individuals who come into contact with PC generally have an indication bias of high health needs. While the baseline function of these individuals is important to reflect those needs, future change in those needs (i.e., as addressed by PC) is also important to measure. In total, 27

**Table 1. List of the 27 preliminary QIs reviewed by the expert panel, how they relate to the 6 themes identified in Phase I and which interRAI assessment can be used to generate the QI.**

| Name and brief description of each theme and QIs that relate to that theme | Failure to Improve | Follow-up prevalence |
|---|---|---|
| *1. Access to care: coordination/continuity of care, access to care providers, access to services at the appropriate time* | | |
| Prevalence of emergency department visits[a],[b] | | X |
| Prevalence of hospital admissions[a],[b] | | X |
| *2. Patient care: Discussion of preferred setting of death across the illness trajectory, advanced goals/care planning* | | |
| Prevalence of clients feeling that progress is not being made regarding completion of personal goals[b] | | X |
| Prevalence of no advance directives[b] | | X |
| Prevalence of clients feeling a lack of completion of financial, legal and other formal responsibilities[b] | | X |
| *3. Caregiver support: How to cope with distress/burden/loneliness, education for caregivers/knowledge to keep client at home, caregiver supports (networks, respite)* | | |
| Prevalence of caregiver distress[a],[b] | | X |
| *4. Symptom management: Treating symptoms and also having a patient-centred approach to care* | | |
| Prevalence of falls[a],[b] | | X |
| Prevalence of severe or excruciating daily pain[a],[b] | | X |
| Prevalence of pain that is not controlled by medications[a],[b] | | X |
| Failure for pain to improve[a],[b] | X | |
| Prevalence of constipation[a],[b] | | X |
| Prevalence of shortness of breath at rest[a],[b] | | X |
| Prevalence of shortness of breath upon exertion[a],[b] | | X |
| Failure for shortness of breath to improve[a],[b] | X | |
| Prevalence of stasis/pressure ulcers[a],[b] | | X |
| Prevalence of a delirium-like syndrome[a],[b] | | X |
| Prevalence of nausea or vomiting[a],[b] | | X |
| Prevalence of fatigue[a],[b] | | X |
| Prevalence of sleep problems[a],[b] | | X |
| Prevalence of poor self-reported health[a],[b] | | X |
| Prevalence of negative mood[a],[b] | | X |
| Failure for negative mood to improve[a],[b] | X | |
| Prevalence of declining social activities that causes distress[a] | | X |
| Prevalence of loneliness[a] | | X |
| Prevalence of anxious complaints[a],[b] | | X |
| *5. Spiritual care: Patient and their family should be provided resources and have access to spiritual care* | | |
| Prevalence of struggling with the meaning of life[b] | | X |
| Prevalence of wanting to die now[b] | | X |
| *6. Home as the preferred place of death: no QIs could be created to address this theme* | | |

[a] indicates a QI that can be generated with the interRAI HC data

[b] indicates a QI that can be generated with the interRAI PC data

preliminary QIs were created for further evaluation in Phase III. Of this list, 20 (74.1%) can be generated with both interRAI instruments, five (18.5%) others can be generated only with the interRAI PC data, and the remaining two QIs, can only be calculated with the interRAI HC data.

## Phase III: Modified Delphi panel to evaluate preliminary QIs

The third phase utilized a modified Delphi panel, based on the RAND/ULCA appropriateness method [57]. The main goal of the Delphi panel was to assess the level of agreement among a group of PC clinicians, researchers, and decision makers with respect to keeping or dropping any of the preliminary QIs. The Delphi method has been widely used in PC research [58]. Prior to the Delphi panel receiving the QI definitions and evaluation criteria, an Excel spreadsheet containing each of the QIs and evaluation criteria (along with an information letter/consent form) was shared with two researchers on our team, and a group of graduate students, for their feedback. We first consulted with two researchers with expertise in the Delphi process and a strong understanding of the goals of our study and a small group of graduate students (roughly 15–20), representing "non-experts," to ensure that the materials were clear and the instructions were easy to follow. The students represented a mix of master's and PhD level students who were all completing degrees within the School of Public Health Sciences at the University of Waterloo.

Decision makers who took part in an interview or focus group, during Phase I, were eligible to participate in the Delphi panel. Panel participants provided written informed consent prior to their participation. They were asked to evaluate each QI on four criteria: 1) *Importance-* the extent to which the indicator reflects an important outcome or issue for those receiving PC or their caregivers; 2) *Validity-* the degree to which the indicator truly captures some aspect of the quality of care (at a population level, not for an individual); 3) *Evidence of improved outcomes-* evidence that improvement in the indicator can have a positive effect on the individual; and 4) *Usability-* the extent to which the QI can be readily interpreted and used to improve care delivery. These criteria were based on previous research [59] and were rated on a scale of 1 to 9 (1 = low; 9 = high) as per the RAND/UCLA method. The evaluation spreadsheet also asked for input on the definition of the numerator/denominator. Finally, participants were given the opportunity to provide open-ended feedback on each QI, as well as space to suggest any additional QIs that the Delphi panelists felt were missing from the list, regardless of whether they could be measured with existing interRAI data.

Participants were given six weeks to complete the documents. If documents were not returned two weeks after the initial deadline, an individual reminder e-mail was sent, asking them to be returned within two weeks. After that time, if they still were not returned, a final reminder phone call was made. Any documents not returned then were considered a non-response and no further contact was made with the participant.

## Determining consensus among raters

As outlined in the RAND/UCLA manual [57], the process to determine the level of agreement among the panel members involves multiple steps.

**Step 1**: focused on determining if there was "disagreement" or "agreement," for each of the four criteria. This involved several calculations in order to arrive at the value of the interpercentile range adjusted for asymmetry (IPRAS). This method is ideal for panel sizes larger than 15, and therefore appropriate for our panel (n = 21) [57]. An example has been provided (S1 File) which outlines how each of the four criteria were determined to have agreement/disagreement.

**Step 2**: involved assessing the value of the median, for each of the four criteria, in conjunction with whether there was "agreement" or not from Step 1. Each of the four criteria were assigned into one of three mutually exclusive groups, namely "discard," "retain," or "review," based on the work of Nakazawa et al. [60]. "Discard" was defined when the

median value was between 1–3 AND there was agreement in Step 1. "Retain" was defined when the median was between 7–9 AND there was agreement from Step 1. "Review" was defined when the median was between 4–6 OR the median was another value AND there was disagreement in Step 1.

**Step 3**: In this step, each QI was assigned into one of three groups, namely "discard," "retain," or "review", based on a review of the rating of the individual QI's four criteria. For example, if any of the four criteria were rated "discard" in Step 2, then the QI would be discarded. If three or four of the criteria were considered "retain" then that QI was kept. Two scenarios were used to decide if a QI should be "reviewed." First, if two of the criteria were considered "retain," then we retained the QI for further review. Second, if two or more of the criteria were considered "review" then the QI would also fall into the "review" category. It should be noted that the research team utilized the Delphi results as a guide, to support decision-making, but the team also used their discretion when making the final decisions about whether or not to keep a QI for further consideration.

## Results

A total of 33 individuals were invited to take part in the Delphi panel. They represented members of our research team (n = 12) and other experts who they suggested that we approach (n = 21). Of the 33 evaluation spreadsheets sent to the Delphi panel members, 21 were completed for a response rate of 63.6%. This group of 21 participants included three individuals who also provided input during Phase I. Among those who did not respond, one person felt that they did not have the necessary expertise to complete the evaluations, and the remainder (n = 11), did not respond after repeated emails. There were six individuals who consented, completed the demographic questionnaire, but then ultimately did not respond. They were very similar to respondents in terms of age (mean = 53.8; sd = 5.9), gender (female = 66.7%) and years of experience working in PC (>10 years = 66.7%). These individuals came from a variety of backgrounds including research (n = 3), nursing (2), and medicine (1).

The Delphi participants were mostly female (71.4%), with a mean age of 46.3 years (sd = 7.3) and the majority had been working in the area of PC for more than 10 years (66.7%; Table 2). The largest proportion (38.1%) came from Ontario, but there were also representatives from British Columbia, Alberta, Nova Scotia and Yukon Territory and two experts from outside of Canada. The majority had a clinical background in nursing or medicine (61.9%), with the remainder involved in PC research or working in the field in the role of a health care administrator or policy maker.

Of the 27 preliminary QIs that were evaluated, 20 (74.1%) were classified as "retain" and the remainder, as "review" (Table 3). None of the proposed QIs had scores that would put them into the "discard" category. The team decided to keep all QIs where the Delphi panel suggested that the QI be retained. Since there were only seven QIs classified as "review," a second Delphi panel was deemed unnecessary. Instead, an online meeting was held with the research team, who provided feedback on these QIs. The final decision was made to drop five of these QIs from further consideration, mainly based on the concern about whether these QIs were truly capturing quality of PC services. Panel members also provided suggestions for new QIs that the team should consider. These included topics such as the timely access to PC services, satisfaction with care, the place of death/preferred place of death, and details around the treatment for certain issues (e.g., for depression, for anxiety). Since there were no interRAI data elements to capture these suggestions, no additional QIs were developed.

**Table 2. Demographic characteristics of individuals who participated in the expert panel.**

|  | Total sample (n = 21) |
|---|---|
| Mean age in years (standard deviation) | 46.3 (7.3) |
|  | % (n) |
| **Gender** | |
| Female | 71.4 (15) |
| Male | 28.6 (6) |
| **Professional background**[a] | |
| Director/Senior Director/Project Lead | 28.6 (6) |
| Physician | 23.8 (5) |
| Registered practical nurse | 19.0 (4) |
| Researcher | 19.0 (4) |
| Other | 14.3 (3) |
| **Years of experience working in palliative care** | |
| <1 year | 4.8 (1) |
| 1–10 years | 28.6 (6) |
| >10 years | 66.7 (14) |
| **Highest degree of education completed** | |
| University—graduate degree | 90.5 (19) |
| College/Undergraduate university degree | 9.5 (2) |
| **Geographic location** | |
| Ontario | 38.1 (8) |
| British Columbia | 19.0 (4) |
| Nova Scotia | 14.3 (3) |
| Alberta | 9.5 (2) |
| Yukon Territory | 9.5 (2) |
| Outside of Canada | 9.5 (2) |

[a]These groups are not mutually exclusive as participants were able to select all that applied

The modified Delphi panel resulted in 22 QIs kept for further testing and validation. Within the QIs capturing clinical issues, the indicators with the highest scores related to importance were those related to pain, shortness of breath and delirium. Those with the highest importance scores in the "psychosocial" area included QIs capturing caregiver distress, mood and loneliness. Three other QIs were kept related to hospital or ED use and advance directives.

## Discussion

To our knowledge, this is the first project to recommend a set of standardized QIs for community-based PC using existing interRAI data. The proposed set of 22 QIs was developed through a rigorous and multi-year process involving many stakeholders and researchers from across Canada and in two other countries. The QIs explicitly capture the issues cited as important by those receiving PC, their families, and those working in the field. The proposed QIs can be measured with existing interRAI instruments, currently used in more than 30 countries around the world. This allows for cross-country comparisons, which have previously been completed using the QIs for nursing homes [61]. Using the existing interRAI data is an efficient and cost-effective use of this information and avoids the additional effort that would be required if quality was assessed by using detailed chart reviews [37] or additional surveying of staff, individuals receiving PC, and families.

**Table 3. Summary of scores from the expert panel and final decision for each of the proposed QIs.**

| Quality indicator[a] | Median score from the expert panel 1 = low and 9 = high (and results from Step 1)[b] | | | | Panel Decision | Final Decision |
|---|---|---|---|---|---|---|
| | Importance | Validity | Evidence of improved outcomes | Usability | | |
| *Prevalence of falls* | 8 | 6 | 8 | 7 | retain | KEEP |
| *Prevalence of severe or excruciating daily pain* | 9 | 8 | 9 | 8 | retain | KEEP |
| *Prevalence of pain that is not controlled by medications* | 9 | 7 | 8 | 8 | retain | KEEP |
| *Failure for pain to improve* | 8 | 7 | 7 | 7 | retain | KEEP |
| *Prevalence of constipation* | 8 | 7 | 8 | 8 | retain | KEEP |
| *Prevalence of shortness of breath at rest* | 9 | 7.5 | 8 | 8 | retain | KEEP |
| *Prevalence of shortness of breath upon exertion* | 7 | 6 | 8 | 7 | retain | KEEP |
| *Failure for shortness of breath to improve* | 8 | 7 | 9 | 7 | retain | KEEP |
| *Prevalence of stasis/pressure ulcers* | 8 | 8 | 8 | 8 | retain | KEEP |
| *Prevalence of a delirium-like syndrome* | 9 | 7 | 8 | 7 | retain | KEEP |
| *Prevalence of nausea or vomiting* | 8 | 7 | 8 | 7.5 | retain | KEEP |
| *Prevalence of fatigue* | 7 | 6 | 8 | 7 | retain | KEEP |
| *Prevalence of sleep problems* | 8 | 7 | 8 | 7 | retain | KEEP |
| Prevalence of poor self-reported health | 6 | 4 | 6 | 5 | review | DROP |
| *Prevalence of negative mood* | 8 | 6 | 8 | 7 | retain | KEEP |
| *Failure for negative mood to improve* | 8 | 7 | 8 | 8 | retain | KEEP |
| Prevalence of declining social activities that causes distress | 7 | 6 | 7 | 6 | review | KEEP |
| *Prevalence of loneliness* | 8 | 6 (D) | 8 | 7 | retain | KEEP |
| *Prevalence of caregiver distress* | 9 | 8 | 8 | 8 | retain | KEEP |
| Prevalence of anxious complaints | 8 | 6 | 8 | 6 | review | KEEP |
| Prevalence of struggling with the meaning of life | 6.5 | 5 (D) | 6 | 5.5 | review | DROP |
| Prevalence of clients feeling a lack of completion of financial, legal and other formal responsibilities | 7 | 5.5 | 7 | 6 | review | DROP |
| Prevalence of clients feeling that progress is not being made regarding completion of personal goals | 7 | 5.5 | 6 | 6 | review | DROP |
| Prevalence of wanting to die now | 8 | 5 | 5.5 (D) | 5 (D) | review | DROP |
| *Prevalence of emergency department visit* | 7 | 7 | 7 | 7 | retain | KEEP |
| *Prevalence of hospital admissions* | 7 | 7 | 7 | 7 | retain | KEEP |
| *Prevalence of no advance directives* | 8 | 7.5 | 7.5 | 8 | retain | KEEP |

[a] The final list of 22 quality indicators that were kept are shown in italicised font

[b] All criteria had "agreement" following step 1 except those marked with "D" to indicate disagreement

While it is important to understand the validity of individual indicators, it is also important to evaluate the content validity of the set of QIs. Several criteria have been proposed with which to carry out this evaluation [62]. For example, it is important that the QIs adequately cover the depth and breadth of the content of interest. One way to assess this criterion is to compare our QI set to the six key themes which were discussed during Phase I. In five of these themes, we were able to develop at least one QI in each theme. However, we could not create any QIs related to theme six, namely, the home as the preferred place of death, as we lack the data elements in the interRAI tools to capture this.

A second criterion of content validity relates to proportional representation, or the number of QIs in each domain that matches the importance of that domain in the assessment of PC quality. We treated each of the six themes as equally relevant since we had no other information with which to judge the importance of these themes. However, it is clear that given the

type of clinical data captured within the interRAI assessments, it was easier to create QIs to address theme four, symptom management, versus the other themes. Another criterion is the costs of measurement which relates to the burden of data collection on providers. In this regard, these preliminary QIs have a very low cost since they are calculated using existing data and no additional data collection is required. The QIs were rated, during the Delphi panel, in terms of importance, providing insight into the criterion that captures the priority or ranking of the QIs. We feel that the QI list does not include redundant QIs, another important criterion, since none of the QIs were suggested to be dropped from the original list during the Delphi process. This provides evidence that all of the 27 preliminary indicators were rated high enough to warrant further consideration. Finally, the QI development did a very good job in addressing stakeholder involvement. The QIs were developed with input from multiple stakeholders which improves the confidence that the QIs have content validity. We were, however, limited to input from only one individual, with lived experience, who was receiving PC, despite nearly a year of effort in attempting to recruit other care recipients from across Canada.

Although the interRAI data represent a very rich source of information, the study team was limited to creating QIs that could be measured using existing items within two interRAI assessments and were unable to create QIs to reflect some salient issues mentioned during the qualitative interviews, many of which have also been cited in the literature as important aspects when assessing the quality of PC. For example, we were unable to create QIs to assess issues related to communication between the person, their family and members of the health care team [63], the use of hospice/PC services [64], the extent to which the person's wishes were met [64], and access to resources and services. We were also unable to determine what specific treatments were received for certain clinical issues like depression and anxiety [18] and how satisfied people were with the PC services they received [65]. This information is important to assess as part of ongoing quality improvement efforts, although it was not directly germane to the clinical rationale for the interRAI assessments, which is care planning. As a result, this type of information would have to be captured using alternative means.

The proposed QIs provide an efficient means to capture key quality issues using existing interRAI data. Since the interRAI assessments are used widely in Canada, and in multiple other countries, these data provide a cost-effective source of information for testing and validating QIs. Although their main function is to assess the overall care needs of the individual, in order to drive care planning, interRAI assessments are useful for case-mix measurement [66–69] and quality assessment [39, 70–72]. In Canada, public reporting on select QIs already exists for long-term care homes [73], and those receiving PC and their families deserve a similar level of transparency. The proposed QIs can contribute to improvements in quality by providing detailed information to individual care providers (e.g., home care agencies) to drive internal continuous quality improvement efforts. The QIs can also provide a solid basis for quality benchmarking and learning, when organizations are compared at a systems-level. Finally, the QI data can be used to educate consumers and to guide health care policy.

The next steps in this project will involve analyzing the existing interRAI data to understand the properties of these QIs (e.g., magnitude of the issue, level of variation between geographic regions) as part of the ongoing validation process to decide which ones should be kept in the final list. Our team has access to approximately 3.7 million home care assessments from five provinces and one territory [50]. In addition, the study team has access to approximately 110,000 interRAI PC assessments, which are completed with palliative home care clients in Ontario only. We also plan to create client-level risk adjusters for each of the proposed QIs to account for differences in risk factors across patient populations [74–76]. This step is very important since complex illnesses, multiple co-existing conditions and case-mix differences can influence the QI measures, irrespective of quality. Organizations that provide care to more

impaired individuals will tend to have higher unadjusted rates, regardless of the quality of care they provide [74]. Risk adjustment methods are therefore needed to maximize the ability to make fair comparisons between providers [77].

As a result of this work, we have identified a set of 22 validated palliative care QIs capturing multiple issues that are important to individuals receiving PC, families and decision makers. This work fills an important gap as many other sectors of the health care system in Canada have access to interRAI-based QIs to assist in decision support and quality improvement [70, 71, 78, 79], but this has been lacking in the PC sector. Once the QIs are finalized, they can be readily embedded into existing software systems for use by Canadian provinces and health authorities who are using the interRAI assessments. They can also be calculated in other countries using these interRAI tools (e.g., the 21 countries using the home care instrument). The final set of QIs will be useful for the purposes of benchmarking performance across different subpopulations of interest, such as health planning/funding regions. The QIs will also provide community-based PC providers, and health system and policy decision makers, with real-time data to support them in targeting their quality improvement efforts and evaluating client outcomes.

## Supporting information

**S1 Table. Operational definitions for each of the 27 quality indicators (QIs).**
(DOCX)

**S1 File. An example of applying the interpercentile range adjusted for asymmetry (IPRAS).**
(DOCX)

## Acknowledgments

The authors gratefully acknowledge Kate Fillmore, Laurel Gillespie, Christina Lawand and Michelle Peterson Fraser for their contributions to this project.

## Author Contributions

**Conceptualization:** Dawn M. Guthrie.

**Data curation:** Nicole Williams.

**Formal analysis:** Dawn M. Guthrie, Nicole Williams.

**Funding acquisition:** Dawn M. Guthrie.

**Investigation:** Dawn M. Guthrie.

**Methodology:** Dawn M. Guthrie, Nicole Williams.

**Project administration:** Dawn M. Guthrie, Nicole Williams.

**Software:** Nicole Williams.

**Supervision:** Dawn M. Guthrie.

**Validation:** Dawn M. Guthrie, Nicole Williams.

**Visualization:** Dawn M. Guthrie.

**Writing – original draft:** Dawn M. Guthrie.

**Writing – review & editing:** Dawn M. Guthrie, Nicole Williams, Cheryl Beach, Emma Buzath, Joachim Cohen, Anja Declercq, Kathryn Fisher, Brant E. Fries, Donna Goodridge, Kirsten Hermans, John P. Hirdes, Hsien Seow, Maria Silveira, Aynharan Sinnarajah, Susan Stevens, Peter Tanuseputro, Deanne Taylor, Christina Vadeboncoeur, Tracy Lyn Wityk Martin.

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
