## [Decision Letter · Decision Letter 0]

7 Jan 2022

PONE-D-21-34279A multi-stage process to develop quality indicators for community-based palliative care using interRAI dataPLOS ONE

Dear Dr. Guthrie,

Thank you for submitting your manuscript to PLOS ONE. After careful consideration, we feel that it has merit but does not fully meet PLOS ONE’s publication criteria as it currently stands. Therefore, we invite you to submit a revised version of the manuscript that addresses the points raised during the review process. 

In addition to responding to the reviewer comments, I do have some significant concerns of my own that need addressing that link into data sharing and participant consent / ethics.  1. In your data sharing statement you state that the secondary data are available by placing a request with the Canadian Institute for Health Information, and that the data do not belong to the researchers. Can you please clarify exactly what data that you used in this study is held with this organisation? It's currently unclear how you collected data from the Delphi panel participants, which you appeared to convene for the purpose of this study, without having control of the data. It's also unclear why this anonymised data could not be included with the submitted paper as per the journal's policy on data sharing.  2. Linked to this is your statement that consent was not obtained from participants because you only used secondary anonymised data. Please explain how were data collected for a very specific research project such as this, where you describe recruiting Delphi panel participants, without collecting primary data. To me this appears to be primary data collection that would require participant consent.  3. Finally, you need to provide further details about recruitment methods in the manuscript. Reviewers 2 and 3 both mention the lack of clarity relating to recruitment to some extent, and this needs significantly strengthening beyond what the reviewers have asked for. What were the characteristics of this 'group of graduate students'? Eg how many were there, what was their expertise, and how did they relate to the project / research team beyond this exercise? For the Delphi panel participants, how were they identified? Who recruited them? How many people were approached and declined to participate or did not respond to an invitation (you provide a response rate, but that's based on total panel members not the total number of people approached to be panel members)? Was there any potential for bias (based on their characteristics)? 

We look forward to receiving your revised manuscript.

Kind regards,

Jason Scott

Academic Editor

PLOS ONE

Journal Requirements:

“I have read the journal's policy and the authors of this manuscript have the following competing interests: Dr. A. Sinnarajah has received research grants (last 5 years) on palliative care (as principal investigator or co-investigator), from the Canadian Institutes of Health Research, MSI Foundation, Canadian Cancer Society, Applied Research in Cancer Control, College of Family Physicians of Canada, Choosing Wisely Alberta, Alberta Innovates Health Research, Alberta Cancer Foundation, Alberta Health Services, University of Calgary, Canadian Frailty Network, Alberta Health and Campus Alberta Health Outcomes and Public Health. He has an academic appointment for palliative care research with Queen’s University and Lakeridge Health (currently), and University of Calgary (last 5 years). Lastly, he is/has paid medical administrative positions with Alberta Health Services and Lakeridge Health. The remaining authors declare that no competing interests exist.”

Reviewers' comments:

Reviewer's Responses to Questions

**Comments to the Author**

1. Is the manuscript technically sound, and do the data support the conclusions?

Reviewer #1: Yes

Reviewer #2: Yes

Reviewer #3: Partly

2. Has the statistical analysis been performed appropriately and rigorously? 

Reviewer #1: Yes

Reviewer #2: Yes

Reviewer #3: N/A

3. Have the authors made all data underlying the findings in their manuscript fully available?

Reviewer #1: Yes

Reviewer #2: No

Reviewer #3: No

4. Is the manuscript presented in an intelligible fashion and written in standard English?

Reviewer #1: Yes

Reviewer #2: Yes

Reviewer #3: Yes

5. Review Comments to the Author

Reviewer #1: I found this article very interesting and I am sure that other readers in the field will also find it so. Observations: The Q.Is were explicit although no real comment was made about the lack of expert by experience or service user input here. What is apparent is the nature of the Q.Is that were dropped. Although these were dropped because of Q.I scores they were focused on psycho social elements. The others were all medical which could be expected however, there are close links between both these elements. I am sure you agree. I understand that this would be a focus of the next stage of research? However, having a rational for this and ensuring that this is a key consideration would have reflected your statement of intention to rectify this. As you say, there was only one family member involved and this could be considered a gap.

Reviewer #2: This manuscript is about a set of quality indicators for community-based palliative care by using interRAI data. Although the author successfully described total 22 QIs developed by panel evaluations.

Major

1. The criteria the author used was slightly different from the criteria in the previous literature the author cited (Importance, Scientific acceptability, Usability, and Feasibility). Furthermore, generalizability of the indicator is also one topic of the criteria for QI. How about the generalizability of these indicators outside the interRAI network? Please discuss.

2. Modified Delphi approach usually includes two or more rounds excluding item generation phase. Why did the author perform only one round? Please discuss.

3. How to determine the domain in this study group? That information would also be useful for readers to capture the process of consensus development.

4. The description about panel members were slightly complex to capture. The relationship between 33 Delphi panel members and 30 PCs and other stakeholders was unclear. Who were these 33 members?

5. Disagreement and panel decision about QI seemed to be complex to understand.

5-i. First, according to the Step 2 in this manuscript, three groups (“discard”, “retain”, or “review”) seemed not to be collectively exhaustive. How to deal with the indicator when the median value was between 7-9 and no agreement in STEP1?

5-ii. Next, in STEP3, if the criteria were considered exact two “review” and two “retain”, which category should the QI be in?

5-iii. Third, how to determine “DROP” or “KEEP” in the Table3? This process was not described in Method section.

5-iv. Finally, Table3 only included appropriateness score for each QI. Disagreement/agreement of each category is necessary to understand the panel decision of each QI precisely. It seemed that the words (“review”, “retain”, and “discard”) were used both for the four criteria and for the individual QI, which could make readers confuse.

6. There is no data about the numerator and denominator of each QI. Furthermore, the detail explanation of each QI should be described.

Minor

1. In supplemental file, “Interpercentile range (IPR)= Upper IPR – Lower IPR” is correct? Upper limit IPR?

Reviewer #3: 1. Line 344: The authors write that “the proposed QIs provide a relatively comprehensive picture of the issues that are 345 widely accepted as key metrics when assessing the quality of PC services[61].” What does “comprehensive” mean in this context? I strongly suggest that you critically discuss the content validity not only of single indicators, but also of the indicator set as a whole (see e.g. Schang L, Blotenberg I, Boywitt D. What makes a good quality indicator set? A systematic review of criteria. Int J Qual Health Care. 2021 Jul 31;33(3):mzab107. doi: 10.1093/intqhc/mzab107. PMID: 34282841; PMCID: PMC8325455.)

Specifically, it would be helpful to explicitly report on the degree to which the proposed QI cover the 6 themes of palliative care that you mention at the beginning of your paper, the gaps in your indicator set and whether the proposed QI represent a balanced view of what is important for palliative care patients, e.g. with respect to the 6 themes.

1. Table 1: Why is the distinction between “clinical” and “psychosocial” indicators relevant? I would expect more content-oriented distinctions between indicators, e.g. with respect to the 6 themes of palliative care you mention.

2. While a Delphi study can help to capture judgements by experts, the qualitative reasons put forward for quantitative ratings remain a “black box”. Also, what does the following sentence mean: “It should be noted that the research team utilized the Delphi results as a guide, to support decision-making, but not as a constraint” (line 272 f.)? To enhance transparency of your findings, I strongly suggest reporting for each QI why you dropped or kept them, and how you made that decision.

3. Why did you decide to drop important indicators only because no data was available? If these indicators were important, wouldn’t it be valuable to develop the required data, e.g. expand on existing interRAI assessments?

4. It remains unclear whose quality is to be measured by the indicators – are specific providers accountable for the features measured? Or a regional health system as a whole? Please specify.

5. Please specify what RN (line 114) means – research nurse?

6. What exactly are “knowledge users” (line 160) and why is it important that they did the recruitment of patients and family members?

6. PLOS authors have the option to publish the peer review history of their article (what does this mean?). If published, this will include your full peer review and any attached files.

Reviewer #1: No

Reviewer #2: **Yes: **Atsushi Mizuno

Reviewer #3: No

---

## [Author Response · Author response to Decision Letter 0]

27 Jan 2022

Our responses to the reviewers' comments have been uploaded with our cover letter.

---

## [Decision Letter · Decision Letter 1]

23 Feb 2022

PONE-D-21-34279R1A multi-stage process to develop quality indicators for community-based palliative care using interRAI dataPLOS ONE

Dear Dr. Guthrie,

Thank you for submitting your manuscript to PLOS ONE. After careful consideration, we feel that it has merit but does not fully meet PLOS ONE’s publication criteria as it currently stands. Therefore, we invite you to submit a revised version of the manuscript that addresses the points raised during the review process. Whilst many of the reviewer comments have been addressed, there are some further comments requiring attention. These relate specifically to the processes around collecting, analysing and interpreting the data. 

We look forward to receiving your revised manuscript.

Kind regards,

Jason Scott

Academic Editor

PLOS ONE

Journal Requirements:

Reviewers' comments:

Reviewer's Responses to Questions

**Comments to the Author**

1. If the authors have adequately addressed your comments raised in a previous round of review and you feel that this manuscript is now acceptable for publication, you may indicate that here to bypass the “Comments to the Author” section, enter your conflict of interest statement in the “Confidential to Editor” section, and submit your "Accept" recommendation.

Reviewer #1: All comments have been addressed

Reviewer #2: All comments have been addressed

Reviewer #3: All comments have been addressed

2. Is the manuscript technically sound, and do the data support the conclusions?

Reviewer #1: Yes

Reviewer #2: Partly

Reviewer #3: Yes

3. Has the statistical analysis been performed appropriately and rigorously? 

Reviewer #1: Yes

Reviewer #2: N/A

Reviewer #3: N/A

4. Have the authors made all data underlying the findings in their manuscript fully available?

Reviewer #1: Yes

Reviewer #2: No

Reviewer #3: Yes

5. Is the manuscript presented in an intelligible fashion and written in standard English?

Reviewer #1: Yes

Reviewer #2: Yes

Reviewer #3: Yes

6. Review Comments to the Author

Reviewer #1: Following amendments reviewer concerns have been addressed. It is evident that manuscript authors have considered all comments to some extent. Although accepted, I would still be interested in seeing a clearer statement about ethics. Line 160 identifies KUs recruited those in receipt of PC and their families. Line 161 because it was inappropriate for the research team to do so. Line 165 identifies 8 KUs on the research team? I imagine KUs have stringent ethical probity however, this is not declared and lines 160, 161 and 165 appear to contradict each other.

Reviewer #2: Although I appreciate efforts about point-by-point responses and the comments about the uniqueness and potential usefulness of this indicators for routine and clinical practice, there are still several concerns about the process in this manuscript.

1. As I described, the generalizability of this results should be discussed. First of all, the author did not discuss about actual dataset of interRAI but interRAI data elements (e.g. data components) only. Thus, these indicators by authors are just only agreement of these elements could be useful for palliative care through consensus strategy. Therefore, the author should describe not only advantage of these indicators but also disadvantages such as lack of information compared with the previous literatures. The advantages of these indicators could be more contrasted. For example, “Prevalence of falls” is unique as indicators for palliative care, especially under sub-theme of symptom management.

2. As far as I am correct, the panel members for this unique Delphi manuscript, especially for Phase III seemed to be completely different from Phase I and II. This should be clarified not only in Results section but also in Method section. As it has not still been improved, it could make readers confuse. The selection bias of panel members should also be described in limitation of this study.

3. The word about “domain” and “theme” could be used carefully not to make readers confuse in this manuscript. It would be better to describe how to determine these domains and themes by this study team (including all Phase I, II and III).

Reviewer #3: (No Response)

7. PLOS authors have the option to publish the peer review history of their article (what does this mean?). If published, this will include your full peer review and any attached files.

Reviewer #1: No

Reviewer #2: No

Reviewer #3: No

---

## [Author Response · Author response to Decision Letter 1]

14 Mar 2022

Reviewer #1:

Following amendments reviewer concerns have been addressed. It is evident that manuscript authors have considered all comments to some extent. Although accepted, I would still be interested in seeing a clearer statement about ethics. Line 160 identifies KUs recruited those in receipt of PC and their families. Line 161 because it was inappropriate for the research team to do so. Line 165 identifies 8 KUs on the research team? I imagine KUs have stringent ethical probity however, this is not declared and lines 160, 161 and 165 appear to contradict each other.

We apologize if this was confusing. The 8 KUs on our team acted as a “conduit” to families and individuals receiving palliative care and helped to link us to these individuals. The KUs and our team felt that it would be inappropriate for us, as a research team, to reach out directly to families/care recipients. Instead, the KUs helped to share our project with potential participants who could then contact us IF they were interested. For example, the KUs helped us by posting flyers on their web pages or by emailing study information to potential participants. In this sense, they were not participants themselves, but were simply helping us to identify individuals who were willing to take part in the research. We have modified line 161 and added one new sentence (line 164-165) to help clarify this process.

Reviewer #2:

1. As I described, the generalizability of this results should be discussed. First of all, the author did not discuss about actual dataset of interRAI but interRAI data elements (e.g. data components) only. Thus, these indicators by authors are just only agreement of these elements could be useful for palliative care through consensus strategy. Therefore, the author should describe not only advantage of these indicators but also disadvantages such as lack of information compared with the previous literatures. The advantages of these indicators could be more contrasted. For example, “Prevalence of falls” is unique as indicators for palliative care, especially under sub-theme of symptom management.

 To clarify, the “interRAI dataset” is made up of the data elements within the assessment and nothing else, so the two things are identical. To be more explicit about which QIs have been cited in the literature, but which we could not measure, we have added references to the literature for those specific QIs where we mention them on lines 376-382 (please note that these added references do not show up in the “track changes” in the word document, but include ref #18, 63, 64 and 65).We feel that we have already described the unique advantages of the QIs derived using the interRAI dataset in the discussion section and feel it would make the discussion somewhat repetitious to add further comment on this. 

2. As far as I am correct, the panel members for this unique Delphi manuscript, especially for Phase III seemed to be completely different from Phase I and II. This should be clarified not only in Results section but also in Method section. As it has not still been improved, it could make readers confuse. The selection bias of panel members should also be described in limitation of this study.

 It was possible for decision makers participating in the Delphi panel to have also taken part in an interview/focus group during Phase I. We did not explicitly exclude them from the Delphi panel. We have modified lines 242-243 in the methods, and lines 291-292 in the results, to make this more explicit.

We do not feel that there was selection bias operating in the panel members. We had 6 individuals who consented, provided a demographic questionnaire but then did not complete the Delphi exercise. They were very similar to Delphi panel members in terms of age, gender, years of experience and clinical background. We have added additional text to describe this group (lines 294-298).

3. The word about “domain” and “theme” could be used carefully not to make readers confuse in this manuscript. It would be better to describe how to determine these domains and themes by this study team (including all Phase I, II and III).

 We see these as two distinct concepts. We have used the term “domain” when discussing the QIs and the extent to which the indicators capture important information to assess quality. We found one instance where the term “domain” likely was inappropriate (line 329) and switched it to the word “area” to be more clear.

On the other hand, we have used the term “theme” specifically as it relates to the qualitative analysis of the workshop results, the focus groups, and interviews. We have added some text (line 153) to make it more clear with respect to the analysis of the workshop data. A quick search confirms that the term “theme” has always been used in this context in the manuscript. This is the accepted wording used in qualitative research to highlight a topic of discussion that was raised by multiple participants. It reflects the way in which the key issues raised by participants “emerge” during the qualitative analysis. As such, we feel that these two terms are indeed unique and have been used appropriately throughout the paper.

---

## [Decision Letter · Decision Letter 2]

23 Mar 2022

A multi-stage process to develop quality indicators for community-based palliative care using interRAI data

PONE-D-21-34279R2

Dear Dr. Guthrie,

We’re pleased to inform you that your manuscript has been judged scientifically suitable for publication and will be formally accepted for publication once it meets all outstanding technical requirements.

Kind regards,

Jason Scott

Academic Editor

PLOS ONE

Additional Editor Comments (optional):

Reviewers' comments:

Reviewer's Responses to Questions

**Comments to the Author**

1. If the authors have adequately addressed your comments raised in a previous round of review and you feel that this manuscript is now acceptable for publication, you may indicate that here to bypass the “Comments to the Author” section, enter your conflict of interest statement in the “Confidential to Editor” section, and submit your "Accept" recommendation.

Reviewer #2: All comments have been addressed

Reviewer #3: All comments have been addressed

2. Is the manuscript technically sound, and do the data support the conclusions?

Reviewer #2: Yes

Reviewer #3: Yes

3. Has the statistical analysis been performed appropriately and rigorously? 

Reviewer #2: Yes

Reviewer #3: N/A

4. Have the authors made all data underlying the findings in their manuscript fully available?

Reviewer #2: No

Reviewer #3: Yes

5. Is the manuscript presented in an intelligible fashion and written in standard English?

Reviewer #2: Yes

Reviewer #3: Yes

6. Review Comments to the Author

Reviewer #2: The authors have answered all the comments and improved manuscripts for readers appropriately. As I asked previously, I would recommend the authors to describe how to make "domain" of these quality indicators.

Reviewer #3: (No Response)

7. PLOS authors have the option to publish the peer review history of their article (what does this mean?). If published, this will include your full peer review and any attached files.

Reviewer #2: No

Reviewer #3: No

---

## [Editor Report · Acceptance letter]

29 Mar 2022

PONE-D-21-34279R2 

A multi-stage process to develop quality indicators for community-based palliative care using interRAI data 

Dear Dr. Guthrie:

I'm pleased to inform you that your manuscript has been deemed suitable for publication in PLOS ONE. Congratulations! Your manuscript is now with our production department. 

Kind regards, 

on behalf of

Dr. Jason Scott 

Academic Editor

PLOS ONE